# Enhanced Carbonylation of Photosynthetic and Glycolytic Proteins in Antibiotic Timentin-Treated Tobacco In Vitro Shoot Culture

**DOI:** 10.3390/plants11121572

**Published:** 2022-06-14

**Authors:** Elena Andriūnaitė, Rytis Rugienius, Inga Tamošiūnė, Perttu Haimi, Jurgita Vinskienė, Danas Baniulis

**Affiliations:** Institute of Horticulture, Lithuanian Research Centre for Agriculture and Forestry, Kaunas Str. 30, 54333 Babtai, Kaunas reg., Lithuania; elena.andriunaite@lammc.lt (E.A.); rytis.rugienius@lammc.lt (R.R.); inga.tamosiune@lammc.lt (I.T.); perttu-juhani.haimi@lammc.lt (P.H.); jurgita.vinskiene@lammc.lt (J.V.)

**Keywords:** in vitro stress response, oxidative stress, proteomics, superoxide anion

## Abstract

Antibiotics are used in plant in vitro tissue culture to eliminate microbial contamination or for selection in genetic transformation. Antibiotic timentin has a relatively low cytotoxic effect on plant tissue culture; however, it could induce an enduring growth-inhibiting effect in tobacco in vitro shoot culture that persists after tissue transfer to a medium without antibiotic. The effect is associated with an increase in oxidative stress injury in plant tissues. In this study, we assessed changes of reactive oxygen species accumulation, protein expression, and oxidative protein modification response associated with enduring timentin treatment-induced growth suppression in tobacco (*Nicotiana tabacum* L.) in vitro shoot culture. The study revealed a gradual 1.7 and 1.9-fold increase in superoxide (O_2_^•−^) content at the later phase of the propagation cycle for treatment control (TC) and post-antibiotic treatment (PA) shoots; however, the O_2_^•−^ accumulation pattern was different. For PA shoots, the increase in O_2_^•−^ concentration occurred several days earlier, resulting in 1.2 to 1.4-fold higher O_2_^•−^ concentration compared to TC during the period following the first week of cultivation. Although no protein expression differences were detectable between the TC and PA shoots by two-dimensional electrophoresis, the increase in O_2_^•−^ concentration in PA shoots was associated with a 1.5-fold increase in protein carbonyl modification content after one week of cultivation, and protein carbonylation analysis revealed differential modification of 26 proteoforms involved in the biological processes of photosynthesis and glycolysis. The results imply that the timentin treatment-induced oxidative stress might be implicated in nontranslational cellular redox balance regulation, accelerates the development of senescence of the shoot culture, and contributes to the shoot growth-suppressing effect of antibiotic treatment.

## 1. Introduction

In vitro plant culture is commonly used for the conservation of genetic resources [1], plant propagation [2], or molecular farming [3,4,5], and it is an important tool for plant genetic transformation studies [6,7]. Bacterial overgrowth of plant tissues or the formation of bacterial colonies on a culture medium is a rather common manifestation of microbial contamination with pathogenic species or the non-fastidious proliferation of endophytic bacteria that can be triggered by changes in environmental conditions or plant host physiology. This leads to growth suppression and reduced propagation efficiency of the in vitro culture [8,9,10,11]. Microbial overgrowth could also present a problem in cases of deliberate introduction of bacteria to the in vitro culture, for example, in bactofection-mediated genetic transformation such as application of *Agrobacterium tumefaciens* for gene delivery [12]. An excess proliferation of the *Agrobacterium* could elicit a defense response in plant cells [13,14,15], leading to a detrimental effect on plant tissue growth and the efficiency of genetic transformation [16,17,18].

To eliminate microbial contamination in the plant tissue culture, a variety of antibiotics and application strategies have been adapted, including medium supplementation with a microbial growth-inhibiting concentration [19] or pretreatment with a high dose of antibiotics [20]. Although the treatment is often reported to improve the propagation or regenerative properties of the tissues [21,22,23,24,25], antibiotics could be toxic to plant cells, especially at the higher concentration that is required for effective decontamination, and even short-term exposure to antibiotics can affect plant growth and development, resulting in reduced plant tissue propagation capacity [26,27]. The negative effect of antibiotics depends on the concentration and duration of exposure, which varies for different plant species [28,29].

Cephalosporin- and penicillin-type antibiotics, such as cefotaxime, carbenicillin, or timentin, which are active against gram-negative *Agrobacterium,* are commonly used to control bacterial overgrowth after bactofection [30,31,32]. The relatively low cytotoxic effect of direct exposure to antibiotic timentin on in vitro plant tissues has been described [30,33,34]. However, our previous studies revealed that treatment of tobacco in vitro culture with the antibiotic at sub-cytotoxic levels resulted in a shoot growth suppressing effect that persists after transfer to medium without antibiotic and is associated with increased levels of oxidative lipid peroxidation injury [35,36]. Therefore, the antibiotic-induced enduring growth suppressing effect could have a negative impact on tobacco plant transformation efficiency, plant regeneration, propagation of the tissue culture or subsequent plant rooting and acclimation.

The oxidative lipid injury is related to the accumulation of reactive oxygen species (ROS) under stress conditions [37] and involves free radical-mediated oxidation of unsaturated lipid chain, leading to the formation of a hydroperoxidised lipid and an alkyl radical [38]. Chloroplasts, mitochondria, peroxisomes, endoplasmic reticulum, and the plasma membrane are major ROS generation sites in plant cells [39]. ROS are generated as by-products of photosynthetic reactions, and ROS overproduced in chloroplasts under unfavorable environmental conditions are implicated in signaling and oxidative damage [40]. Chloroplasts produce ROS because of the excess photons trapped in the photosystem II (PSII) and the electron sinks to molecular oxygen via photosystem I (PSI) [37,41]. Excess energy in PSII is transferred to ground-state oxygen (O_2_), facilitating the production of highly reactive singlet oxygen. Electron channeling to O_2_ that occurs at PSI as part of the water-water cycle results in superoxide (O_2_^•−^) which is subsequently converted to hydrogen peroxide (H_2_O_2_) and O_2_ by the enzyme superoxide dismutase (SOD). In addition, the production of H_2_O_2_ is associated with the recycling of glycolate, which is a product of a photorespiration reaction mediated by the oxygenase activity of ribulose-1,5-bisphosphate-carboxylase/oxygenase (Rubisco) [42]. Despite being significant sources of ROS in the cell, the latter two reactions involve the use of reducing equivalents and ultimately play an important role in protection against oxidative damage under a variety of conditions resulting from excess reducing power such as high irradiance, increased temperature, or carbon dioxide (CO_2_) limitation [43,44].

Protein carbonylation occurs when increased metabolic demand produces higher amounts of ROS in the context of inadequate antioxidant mechanisms [45]. It is mainly mediated by highly reactive hydroxyl radicals (HO^•^) generated from O_2_^•−^ and H_2_O_2_ by the transition metal-catalyzed reactions known as the Fenton and Haber–Weiss reactions [46,47]. Carbonylation of protein amino acids is generally associated with permanent loss of function and may lead to the degradation of the damaged proteins [48]. Protein carbonylation has been well characterized and can serve as a marker of protein oxidation during oxidative stress, aging, and diseases [49,50,51]. The most widely used analytical methods to assess protein carbonylation involve carbonyl modification with 2,4-dinitrophenylhydrazine (DNPH) [52]. Carbonyl groups can also be detected by hydrazides coupled with fluorescent labels [53]. Fluorescent hydrazides have been used for qualitative protein carbonyl assessment such as microscopy imaging but also gel-based proteomics [54].

Stress-related phenomena are common in plant in vitro tissue culture [55,56,57]. The generation of ROS and oxidative damage could lead to the accelerated senescence of plant tissues [58]. Considering elevated levels of oxidative lipid injury enduring for at least several propagation cycles in tobacco in vitro shoots as a consequence of timentin treatment [35], it could be presumed that the oxidative stress-mediated acceleration of senescence could be an important factor contributing to the growth-suppressive effect observed in the timentin-treated tobacco shoot culture. Therefore, in the current study, we assessed the accumulation of ROS (O_2_^•−^ and H_2_O_2_) in tobacco shoot tissues during the propagation cycle. Furthermore, a two-dimensional electrophoresis approach was used to investigate timentin treatment-induced protein expression and carbonylation patterns to elucidate specific details of the mechanism involved in shoot response to antibiotic-induced stress.

## 2. Results

### 2.1. ROS Accumulation and Protein Carbonylation in Tobacco In Vitro Shoots

To study the enduring growth suppression effect in tobacco in vitro shoot culture observed after the timentin treatment as described previously [35], the shoots were cultured on a medium supplemented with timentin at a sub-cytotoxic concentration that was previously shown to be effective for *Agrobacterium* elimination after tobacco genetic transformation [33]. The post-antibiotic treatment (PA) experimental group was prepared by cultivating the timentin-treated shoots on medium without antibiotic under the same conditions as the treatment control (TC) shoots for at least one cultivation passage before the analysis.

The accumulation of ROS was assayed at five time points during the tobacco shoot propagation cycle that were selected based on previous observations of the stress-induced oxidative lipid injury variation during the TC and PA shoot growth and culture senescence [35] (Figure 1; Appendix A). During the first week of cultivation, the O_2_^•−^ concentration in the TC and PA shoot tissues was similar (0.83 ± 0.02 and 0.95 ± 0.05 μmol g^−1^ fresh weight (F.W.), respectively) and a gradual increase in O_2_^•−^ content was detected at the later phase of the propagation cycle, resulting in 1.7 and 1.9-fold increase after three weeks of cultivation as compared to day 1 for the two experimental groups, respectively (Figure 1A). It is notable that, for the PA shoots, the increase in the O_2_^•−^ content occurred earlier and was detectable after one week of cultivation; meanwhile, for the TC shoots, such change occurred during the second week of cultivation. As a result, starting at day 7, a 1.2 to 1.4-fold higher O_2_^•−^ concentration was maintained in the PA shoots compared to the TC shoots.

There was no significant difference in H_2_O_2_ content between the TC and PA experimental groups, and little variation of the H_2_O_2_ concentration was detected in the shoot tissues during the propagation cycle (Figure 1B; Appendix A). Significant higher values of 19.6 ± 0.5 and 21.3 ± 0.4 μmol g^−1^ F.W. were observed only during the first week of cultivation for the TC and PA shoots, respectively.

Visualization of ROS accumulation in shoot tissues by histochemical staining re-vealed a similar dye distribution independent of culture growth stage or treatment. Formazan dye accumulation resulting from nitroblue tetrazolium (NBT) reduction was mainly detected in the leaf lamina compared to stems and leaf petioles; meanwhile, 3,3-diaminobenzidine (DAB) accumulation was more concentrated in shoot stems (Appendix B
Figure A1).

The content of protein-bound carbonyls in tobacco shoots was assessed using the DNPH derivatization method. For TC shoots, protein carbonylation levels varied from 4.0 ± 0.1 to 10.1 ± 1.0 nmol mg^−1^ protein and increase was detected during the first week of cultivation (from day 1 to day 7) and at the end of the cultivation cycle (Figure 1C; Appendix A). PA shoots followed a similar pattern of the initial increase during the first week of cultivation, but for the later part of the propagation cycle the carbonyl content remained steady and overall it was larger compared to TC shoots (from 6.6 ± 0.9 to 13.0 ± 0.6 nmol mg^−1^ protein). The largest (~1.5-fold) difference between the two experimental groups was detected after the first and second weeks of the cultivation cycle, which partially corresponded to the variation observed for O_2_^•−^ content.

### 2.2. Differences in Protein Expression Associated with Tobacco Shoot Response to Timentin Treatment or Tissue Culture Senescence

O_2_^•−^ accumulation in tobacco in vitro shoot tissues could be related to processes of tissue culture senescence and oxidative stress response to timentin treatment, both of which might contribute to an increase in oxidative lipid injury observed previously [35]. Proteomics analysis was used to study potential links between the molecular mechanisms of these two processes. To assess timentin-induced changes, shoot samples of TC and PA experimental groups were collected after the first week of the propagation cycle (day 7), when shoots still maintain active growth but the discrete O_2_^•−^ accumulation pattern is starting to emerge (Figure 1) and the most significant difference in oxidative lipid injury has been detected [35]. Conversely, protein expression differences associated with tissue culture senescence were assessed using samples of the PA shoots collected at early and late stages (after one and three weeks of cultivation, respectively) of the propagation cycle.

Following gel matching, gels included an average of 2917 ± 338 (pH 4–7) and 1449 ± 138 (pH 7–10) protein spots per gel (Appendix B
Figure A2; Appendix A). Statistical analysis revealed 22 proteoforms (corresponding to 13 unique proteins) differentially expressed (≥1.5-fold; *p* < 0.01) between the early and late stages of the shoot propagation cycle (Table 1; Appendix A), and no significant differences were detected as a consequence of timentin treatment. Besides, all detected differences resulted from data acquired using the pH 4–7 range of IPG strips, suggesting that the response was mainly limited to cytosolic proteins.

Proteins upregulated at the late phase of the propagation cycle were involved in photosynthesis (chlorophyll A/B binding protein (CAB1, LHCB2.1, and LHCB2.3), PSII subunit PsbP (PSBP-1) and NADH-ubiquinone/plastoquinone oxidoreductase chain 4L (NDHE)), signaling (remorin (AT3G48940)), protein folding (cyclophilin-like protein (CYP38)), and stress response (germin (GER3)) (Appendix A). Meanwhile, an increase in abundance of the 45 kDa proteoform of Rubisco activase (RCA) was detected and the abundance of 47 and 49 kDa homologs of RCA was reduced. Other proteins with reduced abundance at the late phase of the propagation cycle were related to energy metabolism (glyceraldehyde 3-phosphate dehydrogenase (GAPB), fructose-bisphosphate aldolase (FBA2), and carbonic anhydrase (CA1)) or nitrogen metabolism (glutamine synthetase (GLN1-1)). Reduced abundance was also detected for metallochaperone heavy metal-associated protein 31 (CCH) and subunit E1 of proton pump vacuolar adenosine triphosphate (ATP) synthase (TUF).

### 2.3. Timentin-Induced Changes of Tobacco Shoot Protein Carbonylation

Protein labeling with hydrazine-containing fluorescent dyes and two-dimensional gel electrophoresis was used to assess specific proteome carbonylation pattern changes as a consequence of elevated levels of ROS accumulation and protein carbonyl content in PA tobacco in vitro shoots as compared to TC after one week of cultivation (day 7). Following gel matching, an average of 2351 ± 314 protein spots per gel were included in the analysis (Appendix B
Figure A3; Appendix A). Protein abundance-independent changes in carbonyl content were detected for 26 identified proteoforms (≥1.5-fold; *p* < 0.05), of which the majority (22) had increased carbonylation content in PA tobacco shoot samples compared to TC (Table 2; Appendix A). The reduced carbonylation was detected only for glutamate-1-semialdehyde aminotransferase (GSA2) and triosephosphate isomerase (TPI), and a contrasting effect on carbonylation content was detected for two homologs of chlorophyll A/B binding proteins (CAB1 and LHCB2.3).

### 2.4. Functional Interactions of Differentially Expressed or Carbonylated Tobacco Shoot Proteins

Analysis using Arabidopsis homologs and the String database revealed that the majority of the proteins affected by culture senescence or timentin treatment, with the exception only of CCH and remorin (AT3G48940), formed a highly interlinked network mainly related to photosynthesis and glycolysis biological process and involved several metabolic enzymes as well as stress response and signaling proteins (Figure 2; Appendix A). Significant changes in both parameters, protein expression and carbonylation, were detected for four of the proteins (nodes shown in magenta color). The CA1 and one of the RCA proteoforms had reduced expression at the late stage of the propagation cycle and increased carbonylation in response to timentin treatment. Meanwhile, increased abundance was observed for proteoforms related to chlorophyll A/B binding proteins such as LHCB2.3 and CAB1; however, their carbonylation content response was contrasting.

## 3. Discussion

### 3.1. Accumulation of ROS and Protein Expression Patterns Related to Tobacco In Vitro Shoot Culture Senescence

Similar decreases in H_2_O_2_ content were detected in both TC and PA shoots during the first week of cultivation, which could be likely attributed to the local H_2_O_2_ production in response to mechanical injury stress [60,61,62] that occurs during transfer to the fresh medium. In contrast, a steady increase in the concentration of O_2_^•−^ detected for both experimental groups was detected at the late phase of the shoot propagation cycle, and it corresponded well with the previously described increase of lipid peroxidation injury symptoms [35]. Therefore, it could be presumed that O_2_^•−^ accumulation and oxidative damage are both part of the same process involved in the development of oxidative stress symptoms during senescence of plant tissue culture [63,64].

Proteomics analysis revealed a network of proteins differentially expressed as a consequence of shoot culture senescence that was mostly related to photosynthetic and stress response functions (Table 1; Figure 2). Changes in the photosynthetic function of in vitro shoots are likely linked to the variation of exogenous sucrose concentration in the culture medium and changes in CO_2_ content inside the in vitro vessels during the cultivation process. It has been well established that the availability of exogenous sugars and CO_2_ under in vitro conditions results in changes in Rubisco activity, chlorophyll content, or photosynthetic and Calvin cycle enzyme gene expression of in vitro tissue culture [65,66,67,68,69,70].

It is also notable that the abundance of enzymes involved in the dark reaction of photosynthesis and related to carbon metabolism, such as Rubisco activase (RCA), carbonic anhydrase (CA1), fructose-bisphosphate aldolase (FBA2), and glyceraldehyde 3-phosphate dehydrogenase (GAPA-2), was reduced during the late phase of the tobacco shoot propagation cycle. Meanwhile, increased abundance was detected for several proteoforms related to chlorophyll A/B binding protein (CAB1, LHCB2.1 and LHCB2.3), PSII (PSBP1), and NADH-ubiquinone/plastoquinone oxidoreductase (NDHE), indicating suppressed carbon assimilation processes and enhanced activity of the photosynthetic electron chain. Shoot tissue senescence was also associated with an increased abundance of proteins involved in the stabilization of photosynthetic machinery, such as cyclophilin-like protein (CYP38) [71,72]. These results suggest that increased accumulation of superoxide in the shoot tissues is likely associated with enhanced activity of the electron transport chain rather than photorespiration or other biological processes related to carbon assimilation and metabolism.

### 3.2. Enhanced Protein Carbonylation in Timentin-Treated Tobacco In Vitro Shoots

Enhanced protein-bound carbonyl content was detected in PA shoots after one week of cultivation (from day 7 to day 14) (Figure 1C), when the largest difference in lipid peroxidation injury was also detected between the two experimental groups previously [35]. At the same time, a difference in the O_2_^•−^ concentration emerged between the TC (Figure 1A). Proteomic analysis revealed that the timentin-induced elevated levels of ROS accumulation and oxidative injury in tobacco tissues were associated with increased carbonyl modification within the interlinked network of proteins mainly involved in the biological processes of photosynthesis and glycolysis (Table 2; Figure 2). Enhanced carbonylation was detected for targets related to the dark reaction of photosynthesis, including Calvin cycle and photorespiratory carbon oxidation (Rubisco large subunit (RBCL), Rubisco activase (RCA), chaperonin CPN60A, sedoheptulose-1,7-bisphosphatase (SBPase), transketolase (AT3G60750), carbonic anhydrase (CA1), light-induced water oxidation at PSII (Peptidase M41 (VAR2)), and light-harvesting and energy transfer at photochemical reaction centers (chlorophyll A/B-binding protein (LHCB2.3)).

The key photosynthetic enzyme Rubisco catalyzes the CO_2_-fixing reaction [73], and the function requires RCA that releases tightly bound sugar phosphates from the active site and the molecular chaperone CPN60, involved in the folding of RBCL and RCA [74,75]. SBPase and transketolase are essential enzymes of the regenerative phase of the Calvin–Benson cycle in C_3_ plants and limit the rate of carbon fixation, aromatic amino acid and phenylpropanoid synthesis, and subsequently regulate plant growth [76,77,78,79]. Recently, it was shown that chloroplast stroma carbonic anhydrase is not crucial in CO_2_ assimilation, but the tobacco mutants completely lacking it display abnormal development and increased ROS and stromal pH [80]. Peptidase M41 is involved in the repair of PSII following damage incurred during photoinhibition [81]. LHCB proteins are closely associated with PSII and function as light capturing and excitation energy delivering antennas [82], but they are also involved in photoprotection and response to various stresses [83,84]. It is notable that a similar pattern of photosynthetic protein carbonylation, including Rubisco, rubisco-activase, 33-kDa subunit of the oxygen-evolving complex of PSII, and chlorophyll A/B-binding protein, was linked to leaf senescence in Arabidopsis [85]. This suggests that in our study, timentin treatment-induced oxidative stress might accelerate the development of incipient senescence symptoms in the tobacco shoot tissues.

In addition, increased carbonylation levels of catabolic enzymes involved in glycolysis (enolase, phosphoglycerate kinase, fructose-bisphosphate aldolase 2, and glyceraldehyde-3-phosphate dehydrogenase) were detected in timentin-treated shoots. Similar preferential oxidative damage and inhibition of glycolytic enzymes have been observed in germinating seeds of Arabidopsis [86] and yeast under oxidative stress conditions [87,88]. In parallel to the results described with yeast cells, it could be proposed that glycolytic enzyme inhibition might facilitate cellular defense against oxidative stress, as this would increase the flux of glucose equivalents through the pentose phosphate pathway, leading to the generation of reduced nicotinamide adenine dinucleotide phosphate required for antioxidant enzymes such as thioredoxin and glutaredoxin systems [88,89]. On the other hand, glycolytic pathway inhibition could directly contribute to the shoot growth suppressing effect of timentin treatment.

Other proteins with an increase in carbonylation modification in timentin-treated shoots included actin, translation elongation factors, and the ATP synthase subunit, which have previously been shown to be susceptible to oxidation in bacterial, yeast, or animal cells under stress conditions [87,88,90,91]. However, the possible consequences of the damage to these protein functions and subsequent effects of shoot development remain elusive [92].

## 4. Materials and Methods

### 4.1. Tobacco In Vitro Shoot Culture

TC and PA tobacco (*Nicotiana tabacum* L.) shoots were maintained as described previously [35]. Briefly, a solid Murashige-Skoog medium [93], supplemented with 0.75 mg L^−1^ 6-benzylaminopurine, 30 mg L^−1^ sucrose, and 0.8% agar was used for TC shoot cultivation at 25 ± 1 °C using 150 μmol m^−2^ s^−1^ intensity 16 h photoperiod illumination. PA shoots were treated with timentin for 6 months by cultivation on medium supplemented with 250 mg L^−1^ timentin. Afterward, the PA shoots were transferred to the medium without antibiotic and cultivated for at least one culture passage before experimental analysis.

### 4.2. Analysis of ROS Accumulation

Quantitative analysis of O_2_^•−^ accumulation was performed using NBT staining procedure described by Bournonville et al. [94]. Tobacco shoots were vacuum-infiltrated with a staining solution containing 1 mg L^−1^ NBT, 10 mM sodium azide, 50 mM sodium phosphate, pH 7.8 for 2 min and incubated for 30 min at room temperature (RT) in the dark. Chlorophyll pigments were removed by repeated washing with 96% (*v*/*v*) ethanol. Samples were then lyophilized and homogenized with a Mixer Mill MM 400 (Retsch, Haan, Germany) and dissolved in 2 M potassium hydroxide and dimethyl sulfoxide at 1:1 (*v*:*v*) ratio. The mixture was incubated for 5 min at RT and centrifuged at 10,000× *g* for 10 min. Formazan content in the supernatant was measured using absorbance at 580 nm and ε = 12.8 mM^−1^ cm^−1^ [95].

Production of H_2_O_2_ was detected using DAB staining [96]. Tobacco shoots were vacuum-infiltrated with a staining solution containing 1 mg L^−1^ DAB, 10 mM disodium hydrogen phosphate, 0.05% Tween 20 for 2 min and incubated for 4 h at RT in the dark. Chlorophyll pigments were removed by repeated washing with 96% (*v*/*v*) ethanol. Samples were then lyophilized, homogenized, and dissolved in cold 70% perchloric acid as described above. The mixture was incubated on ice for 5 min and centrifuged at 10,000× *g* for 10 min at 4 °C. The absorbance of the supernatant was measured at 450 nm and H_2_O_2_ concentrations were calculated using a standard calibration curve [97].

For localization of ROS accumulation in shoot tissues, chlorophyll pigments were removed by repeatedly treating the samples with ethanol:acetic acid:glycerol (3:1:1, *v*/*v*) solution instead of ethanol after staining with NBT or DAB dyes. Control samples were prepared by incubating shoots in a staining solution without dye.

### 4.3. Protein Extraction

Tobacco shoot samples were flash-frozen in liquid nitrogen and ground to a fine powder. Total cell protein was prepared from 200 mg of frozen sample using the modified phenol extraction method described previously [98]. Frozen tobacco shoot powder was resuspended in 500 μL of an extraction buffer (0.7 M sucrose, 0.1 M potassium chloride, 0.5 M Tris hydrochloride, 50 mM ethylenediaminetetraacetic acid, 1 mM phenylmethylsulfonyl fluoride, 2% β-mercaptoethanol, 2% polyvinylpolypyrrolidone, pH 7.5), an equal volume of Tris-buffered phenol (pH 8.0) was added, and the sample was incubated on a rotary mixer for 30 min at 4 °C. The sample was centrifuged at 15,000× *g* for 10 min at 4 °C, the upper phenolic phase was transferred into a new tube, 500 μL of an extraction buffer was added, and the extraction procedure and phenolic phase separation procedure were repeated as described above.

For samples used in protein expression analysis, the proteins were precipitated ice-cold 0.1 M ammonium acetate in methanol at −20 °C overnight, centrifugated at 15,000× *g* for 10 min at 4 °C, and the pellet was washed twice with 500 μL of the ice-cold methanol followed by a wash with the same volume of ice-cold acetone. For samples used in carbonylation analysis, ethanol:ethyl acetate (1:1, *v*/*v*) was used for precipitation and wash steps to avoid protein damage by impurities of methanol, and the duration of precipitation was reduced to 1 h. The protein pellet was dried in a vacuum centrifuge.

### 4.4. Quantitative Analysis of Protein Carbonylation

Protein carbonyl content was assessed using a modified method described by Levine [99] and Xia [100]. A 100 μg amount of protein was resuspended in 100 µL 6% SDS, 40 mM sodium acetate pH 5, and insoluble protein was separated by centrifugation at 15,000× *g* for 5 min. The supernatant was mixed with an equal volume of 20 mM DNPH in 10% (*v*/*v*) trifluoroacetic acid and incubated for 30 min at RT. To remove unreacted DNPH, an equal volume of 20% (*w*/*v*) trichloroacetic acid was added. After 30 min incubation on ice, the precipitated protein was centrifuged at 15,000× *g* for 10 min and the pellet was washed four times with ice-cold ethanol:ethyl acetate (1:1, *v*/*v*). The resulting labeled protein pellet was resuspended in 6 M guanidine hydrochloride, and carbonyl concentration was estimated at 370 nm using ε = 22,000 mol^−1^ cm^−1^. Protein concentration was measured using the Roti Quant protein assay (Carl-Roth, Karlsruhe, Germany).

### 4.5. Proteomics Analysis

Protein expression analysis was performed using a differential gel electrophoresis procedure as described previously [101]. Samples were dissolved in protein sample buffer (8 M urea, 2 M thiourea, and 4% 3-[(3-cholamidopropyl) dimethylammonio]-1-propanesulfonate). Aliquots of 50 µg were labeled with Cy3 and Cy5 N-hydroxysuccinimide ester (NHS) fluorescent dyes (Lumiprobe GmbH, Hannover, Germany) by adding 300 nmol of dye dissolved in dimethylformamide and incubating for 30 min on ice. The internal standard was labeled with Cy2 NHS dye. The reaction was stopped by quenching with 1 mM lysine for 15 min on ice. Protein samples were mixed to include two samples of biological repeats and one internal standard. Isoelectric focusing (IEF) was performed on 24 cm IPG strips (BioRad, Hercules, CA, USA) using Ettan IPGphor (GE Healthcare, Chicago, IL, USA). A set of two IEF strips covering a linear pH 4–7 and pH 7–10 gradient was used to include the range of dominant cellular protein pI values in the proteome analysis [102,103]. For the second dimension, proteins were separated on 1-mm thick 10–16% poly-acrylamide gradient gels using Ettan DALTsix (GE Healthcare Chicago, IL, USA). Gels were scanned using a fluorescence scanner FLA 9000 (GE Healthcare, Chicago, IL, USA), and protein abundance quantification and statistical evaluation were carried out using DeCyder 2-D Differential Analysis Software, v.7.0 (GE Healthcare, Chicago, IL, USA).

For carbonylated protein analysis using differential gel electrophoresis, samples were prepared as described by Nikolic et al. [104]. Protein pellets were solubilized in 40 mM sodium acetate, 6% SDS pH 5, and sample aliquots of 150 μg were labeled with 0.5 mM CF647DI hydrazide fluorescent dye (Biotium Inc., Hayward, CA, USA) for 30 min at 25 °C with mixing at 700 rpm. To stabilize the formed hydrazone, sodium cyanoborohydride was added to a final concentration of 0.2 M and incubated for 15 min at 25 °C with mixing. To remove the unreacted dye, the derivatized proteins were precipitated with 20% trichloroacetic acid for 1 h on ice. Precipitated proteins were collected by centrifugation at 15,000× *g* for 10 min at 4 °C and the pellets were washed by resuspending in ethanol/ethyl acetate (50% *v*/*v*) and centrifugation at 15,000× *g* for 5 min at 4 °C. The wash procedure was repeated four times. The resulting pellets were resuspended in the protein sample buffer and protein concentrations were determined by the Bradford method. The protein samples were labeled with CF647DI hydrazide and were then labeled with Cy3 NHS ester fluorescent dye (Lumiprobe, Hannover, Germany) as described above. The pooled sample used as the internal standard and for preparative gel was prepared using the same procedure, except the hydrazide dye was omitted in the labeling step. The internal standard was then labeled with Cy2 NHS dye as described above. Protein separation was performed as described above, except IEF strips covering a linear pH 4–7 range corresponding to the dominant pI value of cytosolic proteins [102,103] were used.

For the preparative gels, 500 µg of unlabeled internal standard was mixed with 50 µg of Cy2 labeled internal standard. After the protein separation procedure, the gel was fixed in 50% methanol and 10% acetic acid for 30 min and stored in 10% methanol, 7.5% acetic acid, and 3% glycerol. Excised protein spots were subjected to trypsin digestion according to a method described previously [105]. Peptides were loaded and desalted on a 100 μm × 20 mm Acclaim PepMap C18 trap column and separated on a 75 μm × 150 mm Acclaim PepMap C18 column using an Ultimate3000 RSLC system (Thermo Fisher Scientific, Waltham, MA, USA) coupled to a Maxis G4 Q-TOF mass spectrometer detector with a Captive Spray nano-electrospray ionization source (Bruker, Billerica, MA, USA).

### 4.6. Data Analysis

ROS accumulation measurements were normalized based on the fresh weight of the shoot tissue and protein carbonyl measurements were normalized to the protein content of the DNPH-labeled samples. Mean values were compared between the experimental groups using a one-way analysis of the variance function of Prism v. 3 (GraphPad Software Inc., San Diego, CA, USA) and the significance of differences compared to the control were identified using Tukey Post-Hoc analysis (*p* < 0.05). Data are presented as the mean of at least four independent experiments and the standard error of the mean.

Four biological repeats from two independent experiments were used for the proteomics analysis. The DeCyder software analysis of variance with the false detection rate function was used to identify statistically significant (*p* < 0.01) differences in protein abundance. A threshold value of 1.5-fold was used for the standardized log abundance-based expression ratio.

For carbonylation analysis using two-dimensional electrophoresis, standardized log carbonyl abundance was normalized to total protein abundance. Significant differences of the mean values between antibiotic-treated and control groups were determined using Student’s *t*-test (*p* < 0.05). A threshold value of 1.2-fold was used for the standardized log abundance-based expression ratio. Principal component analysis (PCA) was performed using the XLSTAT statistical software package (Addinsoft, New York, NY, USA) for MS Excel (Microsoft, Redmond, WA, USA).

Protein identification was performed using the MASCOT server (Matrix Science, London, UK) against *N. tabacum* genome database v.1.0 [59] using a threshold value of at least two peptides and a score > 40. Gene ontology terms were assigned using the Pannzer2 server [106], summarized using the ReviGO server [107], and a semantic similarity plot based on the Lin measure was built [108]. Protein interactions were assessed using the String database [109] using a minimum required interaction score > 0.4.

## 5. Conclusions

Our study revealed an earlier onset of O_2_^•−^ accumulation increase in the timentin-treated tobacco in vitro shoots compared with the control, which coincided with the largest difference in oxidative lipid injury reported previously after one week of cultivation [35]. Although the ROS accumulation increase might imply accelerated development of tissue senescence symptoms in the timentin-treated shoot culture, this was not supported by the absence of protein expression differences characteristic of the late phase of the propagation cycle. However, protein carbonyl modification analysis revealed differential oxidative modification of proteins involved in photosynthetic and glycolytic biological processes. This implies that stress-associated nontranslational remodeling of the photosynthetic and glycolytic metabolic pathways potentially implicated in cellular redox balance regulation could accelerate the development of incipient senescence symptoms in the timentin-treated shoots and could contribute to the shoot growth-suppressing effect of the antibiotic treatment.

## Figures and Tables

**Figure 1 plants-11-01572-f001:**
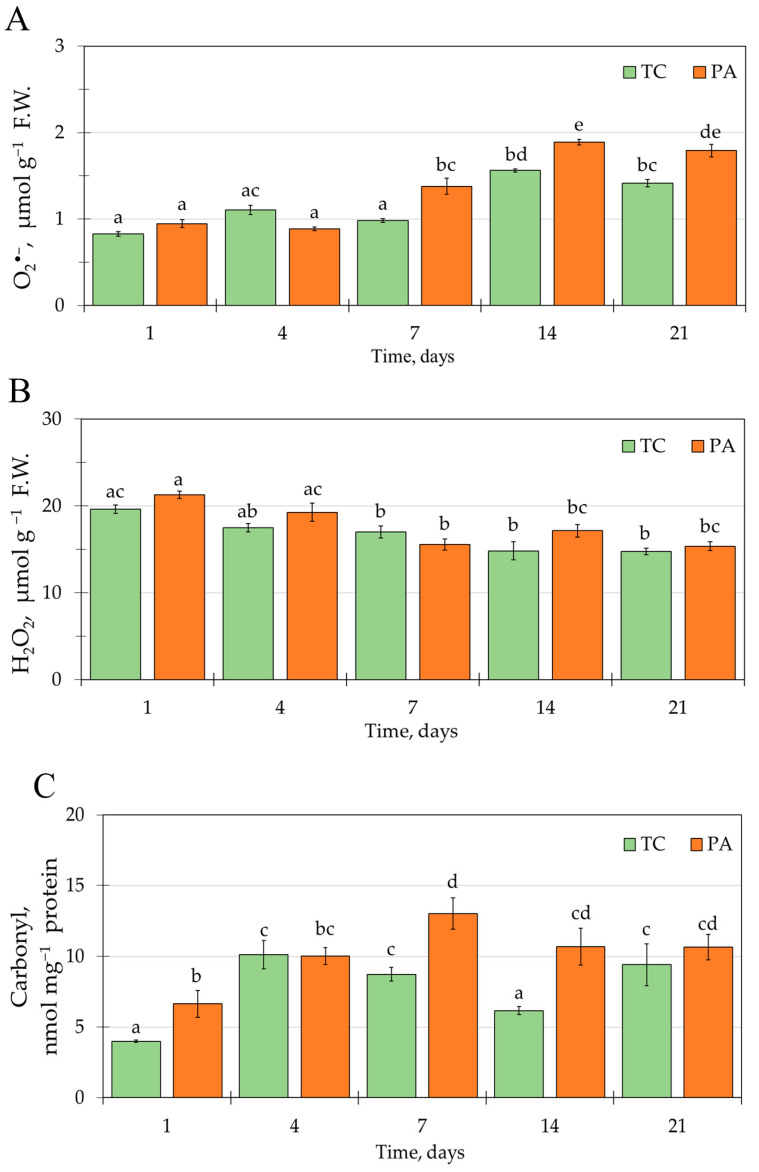
Variation of O_2_^•−^ (**A**), H_2_O_2_ (**B**), and carbonyl (**C**) content in control (TC) and timentin-treated (PA) tobacco shoot tissues during the propagation cycle. Time scale is presented as days after shoot transfer to fresh medium; data are presented as the mean  ±  standard error of the mean; different letters denote significant difference between the mean values (*p* < 0.05).

**Figure 2 plants-11-01572-f002:**
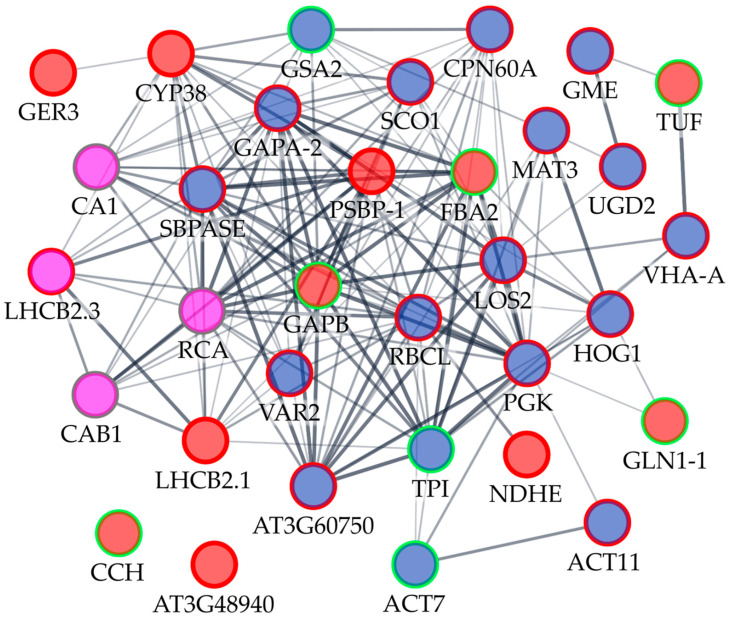
A functional interaction network of tobacco in vitro shoot proteins responsive to culture senescence or timentin treatment. The network was built using the String database using Arabidopsis homologs related to the identified tobacco proteins. Node base color represents proteins differentially expressed between the early and late phase (corresponding to one and three weeks of cultivation, respectively) of the shoot propagation cycle (red), proteins with carbonylation content changes in timentin-treated shoot samples compared to control (blue), or proteins responsive to both factors (magenta). The color of the node border represents a decrease (green), increase (red), or contrasting response (grey) of protein abundance or carbonylation content compared to the control. The thickness of the connecting lines represents the significance of the interaction. Two nodes that lack significant interactions within the network are shown at the bottom.

**Table 1 plants-11-01572-t001:** Proteins differentially expressed between timentin-treated tobacco in vitro shoot samples collected at the early and late stages of the propagation cycle.

No.	Peptide ID ^1^	Protein Name ^1^	TAIR ID	Protein Symbol	Score/P.N./S.C.	M.W./pI	R.A./*p*-Value
1.	0009646g0060.1	Glyceraldehyde 3-phosphate dehydrogenase	AT1G42970.1	GAPB	359/7/25	45.5/8.3	−1.72/0.002
2.	0006485g0040.1	Rubisco activase	AT2G39730.1	RCA	898/18/47	47.9/7.6	−1.75/0.007
3.	0000722g0100.1				466/9/35	49.4/7.5	−1.67/0.002
4.	0000722g0100.1				1303/18/45	49.4/7.5	−2.03/0.001
5.	0023724g0010.1				1136/27/58	47.4/8.2	−1.79/0.001
6.	0000527g0230.1				421/11/36	45.1/7.6	1.74/0.008
7.	0002564g0010.1	Vacuolar ATP synthase subunit E1	AT4G11150.1	TUF	86/2/9	28.9/6.8	−1.81/0.001
8.	0009806g0020.1	Glutamine synthetase	AT5G37600.1	GLN1-1	423/8/20	38.9/5.4	−1.75/0.006
9.	0001317g0050.1	Fructose-bisphosphate aldolase	AT4G38970.1	FBA2	797/12/39	43.7/6.5	−1.56/0.006
10.	0003337g0010.1	Carbonic anhydrase	AT3G01500.2	CA1	617/9/48	32.7/6.7	−1.87/0.001
11.	0007257g0020.1	Heavy metal-associated protein 31	AT3G56240.1	CCH	278/5/34	12.3/4.8	−1.55/0.006
12.	0003337g0100.1	Cyclophilin-like protein	AT3G01480.1	CYP38	549/7/30	42.8/4.7	1.65/0.003
13	0003337g0100.1				646/11/41	42.8/4.7	1.86/0.003
14.	0004193g0010.1	Remorin	AT3G48940.1	AT3G48940	111/5/17	23.3/5.5	2.27/0.004
15.	0002814g0040.1	Chlorophyll A/B binding protein	AT1G29930.1	CAB1	709/13/40	54.1/5.5	1.89/0.001
16.	0002814g0040.1				469/10/37	54.1/5.5	2.12/0.007
17.	0002814g0040.1				576/10/39	54.1/5.5	2.17/0.003
18.	0005511g0010.1		AT2G05100.1	LHCB2.1	677/11/64	28.6/5.5	2.22/0.003
19.	0000441g0070.1		AT3G27690.1	LHCB2.3	187/3/18	28.6/5.5	2.20/0.003
20.	0008321g0040.1	Photosystem II PsbP	AT1G06680.1	PSBP-1	396/8/40	27.1/8.6	2.17/0.005
21.	0004422g0010.1	Germin	AT5G20630.1	GER3	381/5/44	21.5/5.8	2.25/0.001
22.	0001313g0070.1	NADH-ubiquinone/plastoquinone oxidoreductase chain 4L	ATCG01070.1	NDHE	209/3/15	27.3/8.6	1.81/0.005

^1^ The protein ID and name are based on the *N. tabacum* genome database [59]. Abbreviations: TAIR ID—the Arabidopsis Information Resource accession identifier; P.N.—peptide number; R.A.—relative abundance estimated as standardized log abundance-based expression ratio between samples collected at the early and late stage (after one and three weeks of cultivation, respectively) of the propagation cycle; S.C.—sequence coverage; M.W.—molecular weight; pI—isoelectric point.

**Table 2 plants-11-01572-t002:** Proteoforms differentially carbonylated between timentin-treated and control tobacco in vitro shoot samples.

No.	Peptide ID ^1^	Protein Name ^1^	TAIR ID	Protein Symbol	Score/P.N./S.C.	M.W./pI	R.A./*p*-Value
1.	0003983g0050.1	Translation elongation factor EFG related	AT1G62750.1	SCO1	108.4/4/7.6	85.9/5.4	1.5/0.008
2.	0001329g0110.1	Transketolase	AT3G60750.1	AT3G60750	104/3/5	79.8/6.2	1.4/0.035
3.	0002354g0050.1	V-type ATP synthase catalytic subunit alpha	AT1G78900.2	VHA-A	262/8/16	68.6/5.2	1.4/0.009
4.	0023724g0010.1	Rubisco activase	AT2G39730.1	RCA	190/5/17	47.4/8.2	1.8/0.009
5.	0000249g0010.1	Peptidase M41	AT2G30950.1	VAR2	436/12/24	75.6/5.9	1.5/0.024
6.	0004126g0010.1	Chaperonin Cpn60	AT2G28000.1	CPN60A	130/4/8	62.1/5.3	1.5/0.035
7.	0000697g0210.1	S-adenosyl-L-homocysteine hydrolase	AT4G13940.1	HOG1	76/3/6	52.9/6.4	1.5/0.023
8.	0002183g0050.1	Ribulose bisphosphate carboxylase, large subunit	ATCG00490.1	RBCL	43/2/2	50/5.8	1.3/0.034
9.	0002183g0050.1				107/2/4	50/5.8	1.4/0.011
10.	0010922g0010.1	Enolase	AT2G36530.1	LOS2	1003/21/64	47.6/6.5	1.4/0.005
11.	0000565g0160.1	UDP-glucose/GDP- mannose dehydrogenase	AT3G29360.1	UGD2	113/2/6	51.8/6.6	1.2/0.005
12.	0000578g0090.1	S-adenosylmethionine synthetase	AT2G36880.2	MAT3	39/2/3	51.4/6.5	1.4/0.003
13.	0001592g0070.1	NAD-dependent epimerase/dehydratase	AT5G28840.2	GME	206/6/21	42.5/5.9	1.2/0.035
14.	0000029g0330.1	Phosphoglycerate kinase	AT1G79550.2	PGK	49/2/4	42.3/5.7	1.4/0.001
15.	0000369g0010.1	Sedoheptulose-1,7- bisphosphatase	AT3G55800.1	SBPase	609/14/40	44.4/6.1	1.4/0.040
16.	0012115g0030.1	Fructose-bisphosphate aldolase 2	AT4G38970.1	FBA2	410/9/26	42.4/6.1	1.4/0.016
17.	0012714g0020.1				259/7/23	43.4/6.8	1.4/0.004
18.	0010299g0040.1	Glyceraldehyde-3- phosphate dehydrogenase	AT1G12900.1	GAPA-2	1041/23/65	40.8/8.5	1.2/0.020
19.	0003337g0010.1	Carbonic anhydrase	AT3G01500.2	CA1	352/10/41	32.7/6.7	1.7/0.007
20.	0003600g0030.1	Actin-related protein	AT3G12110.1	ACT11	451/9/32	41.7/5.4	1.7/0.039
21.	0002064g0070.1		AT5G09810.1	ACT7	308/9/36	38.9/5.2	−1.5/0.009
22.	0000441g0070.1	Chlorophyll A/B binding protein	AT3G27690.1	LHCB2.3	318/8/45	28.6/5.5	1.42/0.001
23.	0000441g0070.1				219/4/19	28.6/5.5	1.42/0.002
24.	0001434g0050.1		AT1G29930.1	CAB1	41/2/3	30.7/6.4	−1.7/0.026
25.	0011777g0030.1	Glutamate-1-semialdehyde aminotransferase	AT3G48730.1	GSA2	362/8/32	47.7/7.6	−1.4/0.001
26.	0004607g0010.1	Triosephosphate isomerase	AT3G55440.1	TPI	128/4/21	27.2/5.7	−1.7/0.044

^1^ The protein ID and name are based on the *N. tabacum* genome database [59]. Abbreviations: TAIR ID—the Arabidopsis Information Resource accession identifier; P.N.—peptide number; R.A.—relative abundance estimated as standardized log abundance-based expression ratio between the timentin-treated and control samples; S.C.—sequence coverage; M.W.—molecular weight; pI—isoelectric point.

## Data Availability

Not applicable.

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
