# Peer review of "Enhanced Carbonylation of Photosynthetic and Glycolytic Proteins in Antibiotic Timentin-Treated Tobacco In Vitro Shoot Culture"

_plants, 2022, doi:10.3390/plants11121572_

Round 1

Reviewer 1 Report

Antibiotic timentin-induced accumulation of reactive oxygen species, protein expression, and carbonylation response in tobacco in vitro shoot culture 

The authors of the work dealt with the effects of the antibiotic like timentin on in vitro culture of tobacco. The authors showed that the greatest increase in superoxide anion occurs 7 days after transfer of the culture to a medium without antibiotic. Additionally, it was shown that 26 proteins undergo redox modification. 

In my opinion, the work requires in-depth biochemical analyses. Nowadays, the mere determination of carbonylated proteins and spectrophotometric analysis of the superoxide radical anion and hydrogen peroxide are insufficient. 

Also, only spectrophotometric determination of carbonyl proteins after 7 days is unjustified, from my observations the highest level of protein carbonylation is in the first stage of stress and depends not just on the amount of hydrogen peroxide or superoxide radical anion. 

So I would like the authors to generate some additional results concerning spectrophotometric determination of the content of carbonylated proteins during the entire propagation cycle. Also, the determination of protein before determining the carbonyl groups is not entirely justified. The procedure for the determination of carbonyl groups requires pre-purification in ethanol/ethyl acetate and dissolution in guanidine hydrochloride. Protein is lost during these procedures, and not all proteins are dissolved in guanidine hydrochloride. Therefore, the protein level should be measured at the last step in the same solution what carbonylated protein and should be determined at a wavelength of 280 nm. 

Additionally, I would like the authors to measure the activity of the 20S proteasome, which is involved in the removal of carbonylated proteins, and the total level of proteolytic activity. It is also worth measuring the spectrophotometric activity of superoxide dismutase, which is responsible for the superoxide anion dismutation reaction, and the activity of catalase, which is involved in the decomposition of hydrogen peroxide. 

From the technical parts of the article, I would like the authors to include the PCA analysis of obtained protein patterns 

In the table on differentiating or carbonyl proteins, I would like the authors to break down proteins according to UniProt for Go Annotation. 

Please also include p-Values of ratio in the tables. 

Also, the String database needs to be modified because it is very blurry. Protein names can be added in another graphics program and on proteins you can draw arrow indicating if the expression level are up or down. 

I think that additional research will enrich this article and allow for a deeper analysis. 

Author Response

We thank Reviewer for comments and insightful suggestions. Changes described below are indicated by Markup in the uploaded revised manuscript. Line numbers indicated below correspond to the numbers  used in the revised manuscript file including Markup.

Comments and Suggestions for Authors

In my opinion, the work requires in-depth biochemical analyses. Nowadays, the mere determination of carbonylated proteins and spectrophotometric analysis of the superoxide radical anion and hydrogen peroxide are insufficient.

Also, only spectrophotometric determination of carbonyl proteins after 7 days is unjustified, from my observations the highest level of protein carbonylation is in the first stage of stress and depends not just on the amount of hydrogen peroxide or superoxide radical anion. So I would like the authors to generate some additional results concerning spectrophotometric determination of the content of carbonylated proteins during the entire propagation cycle.

The main focus of the study was a proteomics analysis based investigation of the changes that occur in tobacco in vitro shoots after exposure to the antibiotic timentin. Analysis of ROS production together with oxidative lipid and protein injury was used to define the time point during shoot propagation cycle which would represent the physiological differences leading to growth suppression in the timentin-treated shoots. Meanwhile, detail biochemical characterization of stress response variation during the propagation cycle was not an objective of the current study.

It should be noted that experiments described in the current manuscript are directly related to previously published study by TamošiÅ«nÄ— et al. (2022) that was described in the introduction and was referenced in the discussion of the manuscript. Therefore previously published data was also used to interpret the results obtained in the current study. Specifically, day 7 was selected for proteomic analysis partially based on the results of previous study. In the study we showed that reduced shoot biomass accumulation in timentin-treated shoots was associated with increase in oxidative injury of membrane lipids throughout the propagation cycle and the maximum difference was observed on day 7. ROS accumulation and protein carbonyl content analysis results described in the current manuscript were in agreement with the results of the previous study and the data were used to select points relevant for analysis of shoot response to the timentin treatment. We consider that this data provided sufficient basis required for the proteomics based analysis of protein abundance and modification.

However, to address the reviewer request, the quantitative protein carbonylation data is presented in Figure 1C that includes all data points throughout the propagation cycle used for ROS analysis. The results section was modified to describe the results (lines 177-186):

“For TC shoots protein carbonylation levels varied from 4.0 ± 0.1 to 10.1 ± 1.0 nmol mg-1 protein and increase was detected during first week of cultivation (from day 1 to day 7) and at the end of cultivation cycle (Fig. 1C, Supplementary materials Table. S1). PA shoots followed similar pattern of the initial increase during the first week of cultivation but for the later part of the propagation cycle the carbonyl content remained steady and overall it was larger compared to TC shoots (from 6.6 ± 0.9 to 13.0 ± 0.6 nmol mg-1 protein). The largest (~1.5-fold) difference between the two experimental groups was detected after the first and second weeks of cultivation cycle which partially corresponded to variation observed for O2•− content “

Also, the determination of protein before determining the carbonyl groups is not entirely justified. The procedure for the determination of carbonyl groups requires pre-purification in ethanol/ethyl acetate and dissolution in guanidine hydrochloride. Protein is lost during these procedures, and not all proteins are dissolved in guanidine hydrochloride. Therefore, the protein level should be measured at the last step in the same solution what carbonylated protein and should be determined at a wavelength of 280 nm.

Protein content analysis was performed before the carbonylation analysis to adjust protein sample size as well as after DNPH labeling to normalize the carbonyl quantitation data to protein content. As protein absorbance at 280 nm varies for different proteins and the absorbance could potentially vary for samples with different protein composition, protein content was assessed using two methods, 280 nm absorbance and colorimetric RotiQuant Universal assay. Although the results obtained using both methods were similar, eventually the data from the colorimetric assay was used to normalize the carbonyl content and that is indicated in the methods.

The sentence on the lines 501-502 of the Methods section was modified to indicate that the protein content was measured after DNPH labeling:

“ROS accumulation measurements were normalized based on the fresh weight of the shoot tissue, and protein carbonyl measurements were normalized to the protein content of the DNPH-labeled samples.”

Additionally, I would like the authors to measure the activity of the 20S proteasome, which is involved in the removal of carbonylated proteins, and the total level of proteolytic activity. It is also worth measuring the spectrophotometric activity of superoxide dismutase, which is responsible for the superoxide anion dismutation reaction, and the activity of catalase, which is involved in the decomposition of hydrogen peroxide.

The main focus of the manuscript was to assess protein expression and modification changes on day 7 when major increase in oxidative injury was detected and differences in superoxide accumulation occured in the tobacco shoots that undergone timentin treatment. At this growth phase, the proteome analysis did not reveal any changes of protein abundance including proteins involved in regulation of cellular redox homeostasis or protein degradation such as SOD, catalase or 20S proteosome. As activity of the enzymes is mainly regulated by means of gene expression, we considered that the activity measurements would provide little additional information relevant to oxidative stress response interpretation.

From the technical parts of the article, I would like the authors to include the PCA analysis of obtained protein patterns.

The PCA of protein abundance and carbonylation analysis results has been added to the Supplementary materials file Figure S2 and Figure S3, respectively.

In the table on differentiating or carbonyl proteins, I would like the authors to break down proteins according to UniProt for Go Annotation.

Gene onthology terms are presented in the Supplementary materials Table S2 and Table S3. In addition, summary of the GO terms prepared using Revigo server is presented in the Supplementary materials Figures S4 and S5.

Please also include p-Values of ratio in the tables.

p-values were included in Table 1 and Table 2.

Also, the String database needs to be modified because it is very blurry. Protein names can be added in another graphics program and on proteins you can draw arrow indicating if the expression level are up or down.

The image was modified to improve visibility of protein labels.

Reviewer 2 Report

Authors investigated the molecular basis of  timentin treatment-induced growth suppression in tobacco in vitro shoot culture. More precisely, acumulation of ROS and protein oxidative modification - carbonylation. Interestingly, authors demonstrated the mainly glycolytic and photosyntetic proteins were carbonylatied as response to antibiotic treatment. Authors concluded that timentin treatment-induced oxidative stress might accelerate the development of incipient senescence symptoms in the tobacco shoot tissues. 

Major points:

1. Title:

line 187: "no significant differences were detected as a consequence of timentin treatment"

As I can understand imentin did not induce protein expression. The difference was reported between the early and late stages of the shoot propagation cycle. Therefore, please change the title and delete "protein expression". Consider to changethe title to more specific, for example:

Antibiotic timentin-induced accumulation of reactive oxygen species and carbonylation of glycolytic and photosynthetic proteins in tobacco in vitro shoot culture.

2. I cannot find in the manuscript results of carbonylation of proteins analysed between the early and late stages of the shoot propagation cycle. Please include this data. Authors observed differences in protein expression at this stage, therefore oxidative post-translational modifications probably occured beacause ROS content increased earlier in PA.

3. Figures: 

Please check the resolution of images. In pdf file their quality is low. Particularly all the text in figure is blurry.

4. Please explain why the abbreviation PA is used for timentin-treated samples? Usually first letters are used. This abbreviation is not intuitive. I am not against this abbreviation but it would be nice to explain it. Is the TC abbreviation was derived from "treatmenet control"?

Minor points:

l.22compare or comapred?

l.80 abbreviation already introduced in line 78

l.122-124 transfer to methods section

l.130 consider deletion of specific values together with the +/- range, define F.W. otherwise; similar sugeestion for lines 141, 164, 

l.143-144 transfer to discussion

l.162-163  transfer to discussion

l. 181-182 transfer to methods section

l.217-218 transfer to methods section

l. 335 provide full name of NADPH, similarly abbreviations from Methods section shold be explaind, i.e. DMF

l. 520 Consider changing to "Differentially expressed proteoforms" and "Differentially carbonylated proteoforms" in line 526

Author Response

We thank Reviewer for the suggestions on how to improve the manuscript. Changes described below are indicated by Markup in the uploaded revised manuscript. Line numbers indicated below correspond to the numbers  used in the revised manuscript file including Markup.

Major points:

  1. Title:

line 187: "no significant differences were detected as a consequence of timentin treatment"

As I can understand timentin did not induce protein expression. The difference was reported between the early and late stages of the shoot propagation cycle. Therefore, please change the title and delete "protein expression". Consider to change the title to more specific, for example:

Antibiotic timentin-induced accumulation of reactive oxygen species and carbonylation of glycolytic and photosynthetic proteins in tobacco in vitro shoot culture.

We agree that the title might be misleading in relation to the negative results of the protein expression analysis of the timentin-treated shoots. Therefore in the revised manuscript the title was modified based on the reviewer suggestions and to further stress main findings of the study but to keep it concise:

“Enhanced Carbonylation of Photosynthetic and Glycolytic Proteins in Antibiotic Timentin-treated Tobacco In Vitro Shoot Culture”

  1. I cannot find in the manuscript results of carbonylation of proteins analysed between the early and late stages of the shoot propagation cycle. Please include this data. Authors observed differences in protein expression at this stage, therefore oxidative post-translational modifications probably occured beacause ROS content increased earlier in PA.

The carbonylation analysis at the late phase of propagation cycle was not included because the main focus of the study was on the early phase (1 week) where increased superoxide production is associated with the largest increased oxidative lipid injury throughout the propagation cycle (described in TamošiÅ«nÄ— et al. 2022). At this stage the protein expression differences were not detectable, therefore it could be suggested that the timentin-induced enduring growth suppressing effect response could be associated with the enhanced oxidative modification of the proteins involved photosynthesis and glycolysis biological processes.

However, to address the reviewer request, the quantitative protein carbonylation data is presented in Figure 1C that includes all data points throughout the propagation cycle used for ROS analysis. The results section was modified to describe the results (lines 177-186):

“For TC shoots protein carbonylation levels varied from 4.0 ± 0.1 to 10.1 ± 1.0 nmol mg-1 protein and increase was detected during first week of cultivation (from day 1 to day 7) and at the end of cultivation cycle (Fig. 1C, Supplementary materials Table. S1). PA shoots followed similar pattern of the initial increase during the first week of cultivation but for the later part of the propagation cycle the carbonyl content remained steady and overall it was larger compared to TC shoots (from 6.6 ± 0.9 to 13.0 ± 0.6 nmol mg-1 protein). The largest (~1.5-fold) difference between the two experimental groups was detected after the first and second weeks of cultivation cycle which partially corresponded to variation observed for O2•− content “

  1. Figures: Please check the resolution of images. In pdf file their quality is low. Particularly all the text in figure is blurry.

Figures files were prepared and provided in TIFF files at 300 dpi resolution, however the resolution was lost in the PDF version due to conversion to PDF format. PDF converter setting were modified to improve image quality of the revised manuscript. Also please refer to the images in TIFF file provided with submission.

  1. Please explain why the abbreviation PA is used for timentin-treated samples? Usually first letters are used. This abbreviation is not intuitive. I am not against this abbreviation but it would be nice to explain it. Is the TC abbreviation was derived from "treatmenet control"?

Abbreviation “PA” refers to “post-antibiotic treatment” and was used to be consistent with the previous article (TamošiÅ«nÄ— et al. 2022) which describe the enduring antibiotic treatment effect on shoot culture and served as basis for current study extending the research into analysis of the molecular basis of the responce. The “TC” is basically derived from "treatmenet control".

To clarify this issue, the experimental group name abbreviations were further explained in the modified first paragraph of the Results section (lines 128-131):

“Post-antibiotic treatment (PA) experimental group was prepared by cultivating the timentin-treated shoots medium without antibiotic under the same conditions as the control shoots (TC) for at least one cultivation passage before the analysis.”

Minor points:

  1. 22compare or comapred?

Corrected.

  1. 80 abbreviation already introduced in line 78

Corrected.

  1. 122-124 transfer to methods section

The shoot treatment conditions are explained in the Methods section on lines 382-390. However, we feel that including brief description of the tobacco shoot treatment at the beginning of the Results section (lines 128-131) would benefit understanding of the experimental set up of the study. Specifically, there we would like to highlight that the study is related to the residual enduring effect of the timentin treatment but not the response to direct exposure to the antibiotic.

  1. 130 consider deletion of specific values together with the +/- range, define F.W. otherwise; similar sugeestion for lines 141, 164,

We agree that the including specific values might be considered as somewhat redundant with the data presented in the figures, however, omitting the values often leads to the opposite effect when readers are willing to see the specific values as they are not very obvious from the graphic presentation used in figures. Therefore we would prefer to leave the specific values in the text, and the “F.W.” abbreviation was defined on the line 137. 

  1. 143-144 transfer to discussion

The sentence including the interpretation of H2O2 accumulation data was moved to Discussion lines 294-297:

 “Similar decrease in H2O2 content was detected for both, TC and PA shoots, during the first week of cultivation which could be likely attributed to the local H2O2 production in response to mechanical injury stress [59-61] that occurs during transfer to the fresh medium.”

  1. 162-163 transfer to discussion

The sentence referring to the increase in lipid peroxidation injury after one week that was described in previous study was moved to Discussion lines 329-332:

“Enhanced protein-bound carbonyl content was detected in PA shoots after one week of cultivation (from day 7 to day 14) (Fig. 1C) when the largest difference in lipid peroxidation injury was also detected between the two experimental groups previously [35].”

  1. 181-182 transfer to methods section

Sentence was transferred to Methods section lines 461-463.

  1. 217-218 transfer to methods section

Sentence was transferred to Methods section lines 486-488.

  1. 335 provide full name of NADPH, similarly abbreviations from Methods section shold be explaind, i.e. DMF

The abbreviations have been explained through the text. Changes are indicated with Markup in the revised manuscript.

  1. 520 Consider changing to "Differentially expressed proteoforms" and "Differentially carbonylated proteoforms" in line 526

The Figure A2 and A3 titles have been changed according to reviewer suggestion.  

Round 2

Reviewer 1 Report

The authors of the article followed most of the comments. I think the article is suitable for publication in its current form.

Reviewer 2 Report

Manuscript was improved and is ready to be published.